# *Nannochloropsis* sp. Biorefinery: Recovery of Soluble Protein by Membrane Ultrafiltration/Diafiltration

**DOI:** 10.3390/membranes12040401

**Published:** 2022-04-02

**Authors:** Cláudia Ribeiro, Edgar T. Santos, Luís Costa, Carla Brazinha, Pedro Saraiva, João G. Crespo

**Affiliations:** 1LAQV/Requimte, Department of Chemistry, NOVA School of Science and Technology, FCT NOVA, Universidade NOVA de Lisboa, 2829-516 Caparica, Portugal; claudia.ribeiro@algafuel.pt (C.R.); jgc@fct.unl.pt (J.G.C.); 2A4F—Algae for Future, Campus do Lumiar, Estrada do Paço do Lumiar, Edif. E, R/C, 1649-038 Lisboa, Portugal; luis.costa@algafuel.pt; 3CIEPQPF, Chemical Engineering Department, FCT, University of Coimbra, 3030-790 Coimbra, Portugal; pas@eq.uc.pt; 4Dean of NOVA IMS, NOVA University of Lisbon, 1070-312 Lisboa, Portugal

**Keywords:** *Nannochloropsis* sp., protein recovery, circular economy, ultrafiltration, controlled transmembrane pressure, controlled permeate flux

## Abstract

This work proposes a way to maximize the potential of a *Nannochloropsis* sp. biorefinery process, through membrane technology, producing an extract enriched in soluble proteins, free from the insoluble protein fraction, with a low lipid content and eliminating the colored chlorophyll-a. This procedure, following the principles of a circular economy approach, allows for the valorization of a stream from the biorefining of *Nannochloropsis* sp. that, otherwise, would be considered a residue without commercial value. The process proposed minimizes fouling phenomena at the membrane surface, making it possible to achieve high permeate fluxes, thus reducing the need for membrane cleaning and, therefore, contributing to an extended membrane lifetime. Supernatant obtained after centrifugation of a suspension of ruptured *Nannochloropsis* sp. cells was processed by ultrafiltration using a membrane with a cut-off of 100 kDa MWCO. Two different operating approaches were evaluated—controlled transmembrane pressure and controlled permeate flux—under concentration and diafiltration modes. Ultrafiltration operated in a diafiltration mode, under controlled permeate flux conditions, led to the highest soluble protein recovery (78%) with the highest constant permeate flux (12 L·m^−2^·h^−1^) and low membrane fouling.

## 1. Introduction

Proteins are among the microalgae products that have been receiving great attention from the research community. Microalgae proteins are a potential suitable source of high-quality protein not only for human food [1] but also for animal feed [2], and in specific cases, such as the case of phycobiliproteins, for nutraceutical and pharmaceutical applications [3].

Several species of microalgae are known for their high protein contents, with nutritional values similar to other traditional protein sources, such as meat, egg, soybean, and milk [4,5]. Fradique et al. [1] reported having successfully produced a high-protein pasta product through the incorporation of *Chlorella vulgaris* and *Spirulina maxima* in semolina flour. The authors also reported that the cooking quality of pasta was not compromised. Nevertheless, the incorporation of microalgae protein in food products faces several challenges, mostly those related to their organoleptic characteristics, such as odor, taste, and color [6]. Color is the first parameter noticed by the consumer and the common green color, linked to the presence of chlorophyll, leads to very low sensorial acceptance by consumers [7]. Qazi et al. [8] studied the impact on dough rheology and bread quality after enrichment with an ethanol pre-treated *Tetraselmis chuii* protein fraction, to remove the green color and improve the sensory properties. However, despite the clear improvement in dough rheology and bread quality, the authors reported that the green pigmentation was not fully eliminated. For these reasons, microalgae-colored proteins are currently not regarded as valuable products with commercial value. Therefore, in line with the circular economy concept, it is important to find ways to reduce or eliminate the presence of chlorophyll in the microalgae protein.

*Nannochloropsis* sp., which is widely known for the production of omega-3 fatty acids, specifically eicosapentaenoic acid (EPA), also has a significant total protein content (up to 50% of salt-free dry weight) [9,10,11], both soluble (up to 20% of salt-free dry weight) [12] and insoluble. *Nannochloropsis* sp. has a peculiar photosynthetic apparatus, characterized by the presence of only one type of chlorophyll (chlorophyll-a), thus making its removal potentially simple to address [13].

*Nannochloropsis* sp. is a small marine microalga (Eustigmatophyceae). The cells of *Nannochloropsis* sp. are small (3–5 µm), spherical or slightly ovoid [14], with a single chloroplast occupying a large volume of the cell [15], and with a strong cell wall composed of a bilayer structure consisting of a cellulosic inner wall and a hydrophobic algaenan outer layer [16,17]. Algaenan is a resistant biopolymer, and the algaenan layer has been described to be composed of long, straight-chain, saturated aliphatics with ether crosslinks [16,17]. In numerous cases, algaenan is part of a layer called the trilaminar layer structure (TLS), which exhibits two high-electron-density outside layers, sandwiching one layer with a low electron density [18,19]. The resistance of the algaenan cell wall and the small cell diameter confer high mechanical robustness to *Nannochloropsis* sp. cells.

The recovery of soluble proteins from aqueous supernatants, which result from the centrifugation of disrupted biomass, has attracted significant interest. In a microalgae biorefinery, besides the production of *Nannochloropsis* sp. extracts enriched in lipids, a soluble protein fraction can be produced for food applications, free from insoluble proteins and, ideally, without the presence of lipids and colored chlorophyll-a. Lipids tend to be attached to insoluble compounds (e.g., insoluble proteins) that remain in suspension in the aqueous supernatant after centrifugation [20,21]. Thus, it is important to find a suitable process to recover the soluble protein fraction free from the insoluble compounds present (and the lipids attached to them) in the aqueous supernatant obtained by centrifugation. 

Precipitation followed by centrifugation is one of the known methods for protein recovery from aqueous supernatants. Cavonius et al. used a pH-shift precipitation method to obtain a protein isolate from *Nannochloropsis oculata* by lowering the pH to the protein’s isoelectric point [22]. However, this method has significant limitations, such as the low selectivity for different proteins and, in some cases, the irreversible denaturation of proteins.

Alternatively, the use of membrane technologies for fractionation/concentration of proteins from microalgae offers the possibility of operating under mild conditions and thus preventing the denaturation of proteins [23]. Membrane processes, namely the commonly used ultrafiltration or nanofiltration, are pressure-driven processes. The selective transport of the different compounds of a solution is mainly regulated by molecular size-exclusion mechanisms, but also by the Coulombic and hydrophobic interactions between the compounds present and the membrane’s selective top layer [24]. Generally, membrane technologies are economic, as well as easy to operate and to scale-up, although not providing high product selectivity when compared, e.g., with chromatography processes.

Ursu et al. [25] studied the concentration of *Chlorella vulgaris* proteins in the retentate through ultrafiltration (using a membrane with a molecular weight cut-off (MWCO) of 300 kDa) in a concentration mode under fixed transmembrane pressure (fixed at 1.5 bar). The majority of the proteins remained in the retentate (87% *w*/*w* for pH 7). Safi et al. [26] focused on obtaining proteins in the permeate instead of the retentate, free from polysaccharides and chlorophyll. The author studied two-step ultrafiltration with a polyethersulfone (PES) membrane (concentration followed by diafiltration, using the same membrane). Three PES membranes with different molecular weight cut-offs were tested (300 kDa, 500 kDa, and 1000 kDa) at a fixed TMP (2.07 bar). The best performance was achieved with the 300-kDa membrane, and a protein recovery of 37% (*w*/*w* total proteins in the supernatant) was reported [26]. This low protein recovery could be explained by fouling phenomena, which reduce the permeate flow rate and increase the retention by the membrane.

The severe fouling observed at the start of the constant transmembrane pressure (TMP) ultrafiltration process, due to the often very high initial flux of the clean membrane, may be reduced by applying a milder operation, using a controlled permeate flux strategy, operated under subcritical/sustainable permeate flux conditions [27,28]. Monte et al. studied the harvesting of carotenoid-rich *Dunaliella salina* through ultrafiltration in a concentration mode under controlled flux conditions [29]. Similarly, Serra et al. combined ultrafiltration in a diafiltration mode with controlled flux conditions for the purification of viscous arabinoxylans from corn fiber through dia-ultrafiltration [30].

In the present work, membrane fouling occurring during protein filtration could be reduced by operating the ultrafiltration process in a diafiltration mode. The dia-ultrafiltration process enhances the removal of small proteins and carbohydrates, preventing the viscosity of the feed solution from rising over time.

The main objective of this study was to define a suitable methodology for using membrane processing to obtain a high permeate flux with maximum soluble protein recovery (free from chlorophyll-a, insoluble proteins, and lipids). This work also aimed to establish a process with minimum fouling, which could lead to easy membrane cleaning, thereby also contributing to maximizing the membrane lifetime. The global efficiency of the process under study was mainly quantified by the permeate volumetric flux, *J_v_* (L·m^−2^·h^−1^), and by the percentage of soluble protein recovered in the permeate. To our knowledge, ultrafiltration in a diafiltration operation mode under controlled permeate flux has not been reported and evaluated as a methodology for the recovery of soluble proteins in microalgae biorefineries. Besides the study performed by dia-ultrafiltration, ultrafiltration in a concentration operation mode was also tested. In both cases, two different operating conditions were compared, corresponding to controlled transmembrane pressure and controlled permeate flux.

## 2. Materials and Methods

### 2.1. Materials

The *Nannochloropsis* sp. biomass used in this work was cultivated in photobioreactors (PBRs), mechanically disrupted and supplied by A4F—Algae for Future (Portugal). For the ultrafiltration studies, the *Nannochloropsis* sp. disrupted biomass (100 g·L^−1^, on a salt-free dry weight basis—SFDW) was centrifuged at 11,000× *g* (for 2 × 20 min, 5 °C) and the collected supernatant (30 g_SFDW_·L^−1^) was analyzed in terms of its salt-free dry weight, proteins, lipids, and chlorophyll-a content.

Previous studies of ultrafiltration using a hydrophilic ceramic membrane with a nominal molecular weight cut-off (MWCO) of 300 kDa revealed a significant volumetric flux decrease during the process, which may be related to high fouling, namely intrapore fouling (data not shown here). *Nannochloropsis* sp. contains various proteins, which are associated with light-harvesting complexes, with molecular masses ranging between 21 and 32 kDa [31]. Therefore, for optimization of soluble protein recovery, a hollow fiber module containing a tighter ultrafiltration membrane with an MWCO of 100 kDa was used (GE Healthcare, Chicago, IL, USA; model: UFP-100-C-5A). It was composed of 520 hydrophilic polysulfone (PS) fibers, with an internal diameter of 0.5 mm, length of 0.32 m, and total filtration area of 0.2 m^2^. The membrane selected is hydrophilic, to enhance the permeation of soluble proteins, as the extract is aqueous, and to avoid protein denaturation that is prone to occurring at the surface of hydrophobic materials. The membrane-cleaning products Ultrasil 110 and Ultrasil 75 were purchased from Ecolab (Mississauga, Ontario, Canada) and ethanol (97%) was obtained from Panreac (Barcelona, Spain). For analytical procedures, acetone 90.0% and methanol 99.8% were purchased from JMGS (Odivelas, Portugal), chloroform was obtained from Honeywell/Riedel-de Haën (Seelze, Germany), and the Pierce™ BCA Protein Assay Kit was acquired from Thermo Fisher Scientific (Waltham, MA, USA).

### 2.2. Membrane Processing

In this work, the supernatant (30 g_SFDW_·L^−1^) from the centrifugation of *Nannochloropsis* sp. disrupted biomass and its dilution, with a dilution of 1:3 corresponding to 10 g_SFDW_·L^−1^, were processed with the selected membrane system, as summarized in Table 1. The supernatant (30 g_SFDW_·L^−1^ and 60 g_DW_·L^−^^1^) from the centrifugation of *Nannochloropsis* sp. disrupted biomass was diluted with *Nannochloropsis* sp. culture medium, which has a salt concentration of 30 g_DW_·L^−1^, to a dilution ratio of 1:3. After the dilution, the concentration of the diluted supernatant was 10 g_SFDW_·L^−1^ and 40 g_DW_·L^−1^.

Primarily, the membrane ultrafiltration experiments were conducted in a concentration operation mode, under controlled transmembrane pressure (TMP) conditions, as it is the most commonly reported in the literature (see the scheme of membrane setup in Figure 1a). The membrane filtration unit constituted a feed tank (1), recirculation pump (2), pressure gauges (3), membrane module (4), and permeate tank (5). Additionally, the ultrafiltration experiments were also performed in a diafiltration operation mode followed by concentration, under controlled transmembrane pressure conditions (see the scheme of membrane set-up in Figure 1b), where a water tank (6) and a peristaltic pump, for a water feed, were added. The value of transmembrane pressure was set as 0.2 bar (see Table 1) to preserve mild operating conditions.

As shown in Table 1, three experiments were also performed under controlled permeate flux conditions, where the permeate flux was fixed and the TMP was allowed to vary. The membrane setup for the controlled permeate flux experiments was similar to the preceding one (setup for the controlled TMP experiments), both for concentration and diafiltration operation modes (see the scheme of membrane setup in Figure 1c,d, respectively). When operating under controlled permeate flux, its value was imposed by introducing a positive displacement pump (5) in the permeate circuit, which controls and defines the permeate flux, as presented in Figure 1c,d. The value of permeate flux was set at 12 L·m^−2^·h^−1^ (see Table 1). This value was selected after preliminary experiments, which showed that at this flux, no increase of transmembrane pressure occurs (sub-critical, sustainable flux conditions).

In each membrane experiment, the feed was pumped from the feed tank through the membrane module using a peristaltic pump and recirculated to the feed tank, with a cross-flow velocity of 0.25 m·s^−1^. The feed reservoir was filled with 1.5 ± 0.5 L of protein aqueous extract at 20 °C and pH 6.7 ± 0.1. The permeate was collected in the permeate tank and the permeate flux was monitored by mass acquisition using an electronic balance (Kern 572, Kern, Balingen, Germany). For the ultrafiltration in a diafiltration operation mode, the diafiltration volume *D* (-) was calculated through Equation (1): (1)$$D=\frac{V_{water\;added}}{V_{feed}}$$
where *V_water added_* (L) is the volume of the water (diafiltration solvent) added and *V_feed_* (L) is the volume of the feed. For the ultrafiltration process operated in a concentration operation mode, the concentration factor, *CF* (-), was determined according to Equation (2):(2)$$CF=\frac{m_{t0}}{m_{t0}-m_{permeate}}$$
where *m_t_*_0_ (kg) represents the mass in the feed compartment in the beginning of the experiment and *m_permeate_* (kg) is the permeate mass collected during a given period.

The permeate volumetric flux, permeance, apparent rejection of soluble proteins, and percentage of soluble protein recovered in the permeate were calculated during the membrane ultrafiltration experiments. 

The volumetric permeate flux *J_v_* (L·m^−2^·h^−1^) was calculated according to Equation (3):(3)$$J_{v}=\frac{m_{permeate}}{\rho \cdot A\cdot t}$$
where *m_permeate_* (kg) is the mass of permeate, *ρ* (kg·m^−3^) is the density of the permeate, *A* (m^2^) is the total membrane filtration area, and *t* (h) is the time of permeation. The permeance *L_p_* (L·m^−2^·h^−1^·bar^−1^) was calculated through Equation (4):(4)$$Lp=\frac{J_{v}}{TMP}$$
where *J_v_* (L·m^−2^·h^−1^) is the volumetric permeate flux and *TMP* (bar) is the transmembrane pressure. In each test (controlled transmembrane pressure concentration/diafiltration and controlled permeate flux concentration/diafiltration), the global apparent rejection of a target compound *R_i_* (%) and the percentage of soluble protein recovered in the permeate were calculated during the filtration experiments through Equation (5) and Equation (6), respectively:(5)$$R_{i}\left(\%\right)=1-\frac{C_{i,perm}}{C_{i,\;feed}}$$
(6)$$\%\;Recovery_{s.protein}=\frac{m_{protein,\;perm}\;\left(t\right)}{m_{protien,\;feed}\;\left(t_{0}\right)}$$
where in Equation (5), *C_i,perm_* (g·L^−1^) is the concentration of the target compound in the collected permeate and *C_i,feed_* (g·L^−1^) is the concentration of the same compound in the feed (retentate) compartment, and in Equation (6), *m_s.protein,perm_* (*t*) (g) is the mass of soluble protein in the permeate compartment at a defined time of the experiment (*t*) and *m_s.protein,feed_* (*t*_0_) (g) is the mass of soluble protein in the feed compartment at the initial time of the experiment.

The soluble protein loss was defined as the soluble protein that is not recovered in the permeate for each experiment and was evaluated through a soluble protein mass balance, as shown in Equation (7):(7)$$m_{s.protein,\;feed}\;\left(t_{0}\right)=m_{s.protein,\;perm}\;\left(t_{F}\right)+m_{s.protein,\;reten}\;\left(t_{F}\right)+m_{s.protein,\;accum}\;\left(t_{F}\right)\;$$
where *m_s.protein,feed_* (*t*_0_) (g) is the mass of soluble protein in the feed compartment at the initial time of the experiment, *m_s.protein,perm_* (*t_F_*) (g) is the mass of soluble protein recovered in the permeate at the end of the experiment, *m_s.protein,reten_* (*t_F_*) (g) is the mass of soluble protein in the retentate at the end of the experiment, and *m_s.protein,accum_* (*t_F_*) (g) is the soluble protein adsorbed/accumulated on the membrane surface and/or pores (surface intra-pore fouling).

The fouling phenomena, defined as the undesirable adsorption/deposition of dissolved solutes or suspended particles, was evaluated through the resistance-in-series model [32]:(8)$$J_{v}=\frac{TMP}{\eta \cdot R_{total}}\ll \gg R_{total}=\frac{TMP}{\eta \cdot J_{v}}$$
where in Equation (8), *J_v_* (m·s^−1^) is the volumetric permeate flux, *TMP* (Pa) the transmembrane pressure, ղ (Pa·s) the dynamic viscosity of the permeate (we considered the water viscosity at 20 °C, 1.002 × 10^−3^ Pa·s, since we are dealing with an aqueous extract), and *R_total_* (m^−1^) represents the total resistance. We considered that the total resistance *R_total_* (m^−1^) results from a series of resistances introduced by the intrinsic membrane resistance *R_m_* (m^−1^) and the resistance caused by the fouling, which may comprise two distinct contributions: *R_rev_* (m^−1^), standing for the resistance that disappears when the pressure across the membrane is released and pure water is passed through the system at the end of the experiment, removing unbounded solutes from the membrane surface—i.e., reversible fouling resistance; *R_irrev_* (m^−1^), the resistance that is only removed when a cleaning cycle using chemical agents is applied, removing the compounds chemically bonded to the membrane—i.e., irreversible fouling resistance. Therefore, we have then for the total resistance:(9)$$R_{total}=R_{m}+R_{rev}+R_{irrev}$$

The resistance that the membrane offers to the permeation of pure water, *R_m_* (m^−1^), was calculated through the hydraulic permeance measured at the beginning of each experiment:(10)$$R_{m}=\frac{1}{Lp_{w}\cdot \eta_{w}}$$
where *Lp_w_* (m·Pa^−1^·s^−1^) is the hydraulic permeance and ղ*_w_* (Pa·s) is the viscosity of water at 20 °C (1.002 × 10^−3^ Pa·s).

To estimate the remaining resistances, as mentioned above (9), the total resistance, *R_total_* (m^−1^) was calculated at the end of each *Nannochloropsis* sp. supernatant filtration experiment, using the last measured value of volumetric permeate flux, *J_v_* (m·s^−1^), and *R_m_* + *R_irrev_* (m^−1^) values were obtained through the volumetric flux of water, measured after a flush with pure water (20 °C) to eliminate reversible fouling. By subtracting *R_m_* + *R_irrev_* (m^−1^) from *R_total_* (m^−1^), it was then possible to calculate *R_rev_* (m^−1^) and compute *R_irrev_* (m^−1^), subtracting the known parameters, *R_m_* (m^−1^) and *R_rev_* (m^−1^), from the total resistance *R_total_* (m^−1^).

### 2.3. Membrane Cleaning

To achieve reproducible membrane experiments, after membrane cleaning, the hydraulic permeance was measured. The hydraulic permeance measured with deionized water at 20 °C before the experiments was 221 ± 3 L/(m^2^ h bar). After each trial, the membrane was cleaned, using a pre-established cleaning cycle to recover the initial hydraulic permeance. The cleaning procedure comprised the following steps: use of an alkaline solution (at 45 ± 2 °C) with a concentration of 0.5% (*w*/*w*) of Ultrasil 110 recirculated for 30 min, to remove organic fouling; then, an acid solution of Ultrasil 75 with a concentration of 0.03% (*w*/*w*) was recirculated for 30 min (at 45 ± 2 °C) to remove the inorganic fouling; the last step consisted of circulating ethanol (at 45 ± 2 °C) for 20 min with a concentration of 70% (*v*/*v*), to recover the hydrophilicity of the membrane surface and restore the water permeability [33]. Between these different cleaning operations and steps, the membrane was flushed with deionized water until the pH was stabilized at 7 and negligible conductivity was reached.

### 2.4. Dry Weight and Salinity Quantification

The dry weight of each sample was determined by the weight difference before and after water evaporation. The drying program was run at 105 °C until a constant weight was achieved in a moisture analyzer (Kern DBS 60-3, Kern, Germany). To determine the sample salt-free dry weight (SFDW), a salinity refractometer (Kern ORA 1SA, Kern, Germany) was used.

### 2.5. Protein Quantification

Total protein quantification of the different samples was performed using a Pierce™ BCA Protein Assay Kit (Thermo Scientific, Waltham, MA, USA). The calibration curves and quantification assays were performed according to the supplier’s instructions. Soluble protein quantification was performed after centrifugation (14,000× *g* for 15 min) of each sample to precipitate the membrane proteins that remained attached to cellular debris. The proteins that remained in the supernatant after centrifugation were considered soluble proteins and analyzed through a bicinchoninic acid (BCA) assay [34,35].

### 2.6. Total Lipids Quantification

The total lipid content was extracted with a solvent mixture of 1:1 methanol:chloroform, as described by Ryckebosch et al. [36]. Briefly, 100 mg of previously freeze-dried samples was mixed with 4 mL of methanol. Chloroform (2 mL) and water (0.4 mL) were then added and the sample was vortex-mixed for 45 s. Chloroform (2 mL) and water (2 mL) were added again and the sample was vortex-mixed for another 45 s. After that, the sample was centrifuged (4000× *g* for 10 min). The upper layer was reserved in a clear tube and the remaining pellet was re-extracted with 4 mL of methanol:chloroform 1:1. The centrifugation step was repeated and the solvent layers were passed through a layer of anhydrous sodium sulfate. Afterward, the solvent was removed by rotary evaporation and the lipid content was determined gravimetrically.

### 2.7. Chlorophyll-a Quantification

Quantification of chlorophyll-a was performed through spectrophotometric analysis. Briefly, 1 mg of a freeze-dried sample was mixed with 4 mL 90.0% acetone and vortex-mixed for 60 s. Then, the sample was centrifuged at 4000× *g* (for 10 min, 5 °C). The extracted chlorophyll-a, recovered in the supernatant, was analyzed by spectrophotometry (Genesys 10S UV-VIS), and the chlorophyll-a content was determined according to Equation (11), proposed by Ritchie et al. [37], for two different wavelengths:(11)$$Chl\;a\;\left(\mu g\cdot {\mathrm{mL}}^{-1}\right)=-1.7858\times A_{647}+11.8668\times A_{664}$$

## 3. Results and Discussion

The analysis of the results obtained is focused here on the ultrafiltration process, after centrifugation of disrupted *Nannochloropsis* sp. biomass. In Section 3.1, characterization of the supernatant from the centrifugation of disrupted *Nannochloropsis* sp. biomass is presented. Subsequently, in Section 3.2, results from the experimental work performed with ultrafiltration/diafiltration are presented and discussed.

### 3.1. Nannochloropsis sp. Supernatant Characterization

The disrupted (>90% disintegration) *Nannochloropsis* sp. biomass (100 g_SFDW_·L^−1^) was centrifuged and the supernatant was collected and analyzed in terms of the salt-free dry weight, proteins, lipids, and chlorophyll-a. The supernatant samples, used as feed solutions for filtration experiments without dilution, presented on average a salt-free dry weight of 30.2 ± 0.8 g·L^−1^. The supernatant samples’ characterization (Table 2) revealed that 83% of the proteins (*w*/*w* total proteins) present in the supernatant were soluble. The total lipids analysis shows that the supernatant contained 4.37 ± 0.22 g·L^−1^ lipids, on average. The *Nannochloropsis* sp. used in the present study was grown under N-replete conditions; lipids were mainly present in the form of polar lipids (glycolipids and phospholipids), which are crucial to preserving cell-membrane integrity [20]. These lipid classes are expected to have interactions with cytoskeletal proteins [21] that are responsible for the structural organization of the cell and to ensure several cellular processes (e.g., cell division, motility, and contractility) [38]. Therefore, some of the polar lipids attached to cytoskeletal proteins might remain in the aqueous supernatant.

### 3.2. Ultrafiltration/Diafiltration

During the ultrafiltration experiments, different operation modes (concentration and diafiltration) and operating conditions were compared (controlled transmembrane pressure, TMP, and controlled permeate flux). In an ultrafiltration process under controlled TMP, the transmembrane pressure is fixed and the flux is allowed to evolve, declining as the membrane fouls. On the other hand, ultrafiltration under controlled permeate flux allows for the precise control of convective transport toward the membrane surface, making it possible to adjust the viscous force near the membrane [28,39,40]. If the controlled permeate flux is imposed assuring a sustainable flux (subcritical conditions), membrane fouling is avoided, or at least, minimized. In fact, unlike the controlled permeate flux operation, the controlled transmembrane pressure operation is more prone to causing severe fouling phenomena, with a typical trend for high initial permeate flux, followed by a drastic reduction [27].

The performance of the ultrafiltration/diafiltration process for both operating conditions is presented below.

#### 3.2.1. Volumetric Flux and Permeance

The results for the volumetric flux (*J_v_*) obtained, while maintaining a controlled transmembrane pressure at 0.2 bar or a controlled permeate flux of around 12 L·m^−2^·h^−1^, are presented in Figure 2. The volumetric flux decreases rapidly at the beginning of the process for the ultrafiltration under controlled TMP, from over 8 to 4 L·m^−2^·h^−1^ (the permeate flux decline is extremely fast in the beginning, and after the 5 min required to measure the permeate flux, this value is already significantly lower than its initial clean water flux). This behavior is a consequence of the intensive fouling that occurs at the start of a constant TMP ultrafiltration operation [27,39]. These results are comparable with those obtained by Safi et al. [26], where the permeate flow rate decreased at the beginning of the ultrafiltration process, as measured using a 300 kDa MWCO membrane under a constant TMP of 2.07 bar. The ultrafiltration process under controlled permeate flux (Figure 2B) is known to provide milder operating conditions when compared to the ultrafiltration process under controlled TMP (Figure 2A), allowing a higher value of volumetric flux, at around 12 L·m^−2^·h^−1^, to be maintained during the entire process. The value of 12 L·m^−2^·h^−1^, selected for the imposed permeate flux, was based on previous experience in the processing of biological media (data not shown here).

Similar to the volumetric flux (Figure 2A), the membrane calculated permeance, considering the osmotic pressure difference as negligible, followed the same trends for the controlled TMP experiments (Figure 3A). The selection of an ultrafiltration process under controlled permeate flux provides better results in terms of the process operating performance. Still, it should be noted that even with a gentle permeate flux of 12 L·m^−2^·h^−1^ (experiments 4 and 5), the transmembrane pressure increased due to some fouling, and consequently, the permeance decreased.

#### 3.2.2. Soluble Protein Recovery and Permeate Characterization

The effects of different operation modes (concentration/diafiltration) and operating conditions (controlled TMP and controlled permeate flux) on the recovery of soluble protein in the permeate can be observed in Table 3, as calculated by Equation (6) (see Section 2.2). The experiments under controlled permeate flux exhibit higher values of protein recovery in the permeate (between 44% and 62% for concentration mode and 78% for diafiltration mode) than the experiments conducted under controlled TMP (between 35% and 40% for concentration mode and 58% for diafiltration mode). The lower protein recovery obtained for controlled TMP experiments is related to the more severe fouling associated with these process operating conditions, as it is known that fouling can act in the selectivity/rejection of solutes during the filtration process [27,39].

The results obtained in terms of protein recovery during the diafiltration under controlled TMP are slightly lower than those obtained by Balti et al. [41] for the diafiltration of soluble proteins from Spirulina. The authors reported 64% recovery of soluble proteins in the permeate, using a 150 kDa ceramic membrane at a fixed TMP of 4 bar. It is also important to note that the authors applied two more diavolumes (a total of five) than those applied during the diafiltration described in the present work (a total of three), which might have contributed to a slightly higher protein recovery.

The mass balance for soluble proteins (Table 4) also reinforces the association of lower soluble protein recovery with more severe fouling, due to soluble protein adsorption. The experiments under controlled TMP exhibit higher soluble protein losses due to adsorption (OUT_Adsorbed_/IN_Feed_) to the membrane during the filtration process (between 42 ± 0.4% and 35 ± 1.5% for concentration mode and 37 ± 2.0% for diafiltration mode), than the experiments under controlled permeate flux (between 37 ± 2.7% and 13 ± 2.6% for concentration mode and 7 ± 0.4% for diafiltration mode). These results also confirm that it is easier to clean a membrane after operation under controlled permeate flux than under controlled TMP.

The results presented in Table 3 show that the diafiltration experiments lead to higher soluble protein recovery in the permeate. A comparison of the two diafiltration experiments (Exp.3—under controlled TMP and Exp.6—under controlled permeate flux) is shown in Figure 4. Comparison of these two experiments reveals that the soluble protein recovery is higher among all diafiltration volumes when the experiment is operated under controlled permeate flux (Figure 4A). Figure 4B shows that the diafiltration + concentration under controlled permeate flux presents clear advantages over the diafiltration + concentration under controlled TMP, not only in terms of soluble protein recovery in the permeate (58% for Exp.3—under controlled TMP and 78% for Exp.6—under controlled permeate flux) but also in the reduction of the overall process time.

The characterization of the permeates after each trial shows that, for all the experiments, insoluble proteins, lipids, and chlorophyll-a were retained in the retentate (Table 5), which is comparable with the results obtained in other ultrafiltration studies [25,26]. Polysaccharides may have been retained, according to other studies [26,42], as expected, given their large molecular weights. Chlorophyll-a retention may be explained by its adsorption to small hydrophobic cell debris that remained in the supernatant but was larger than the membrane pores. The results obtained in terms of chlorophyll-a retention are comparable with those reported by Safi et al. [26] when using a 300 kDa PES membrane, operated under a controlled TMP. The authors reported that the chlorophyll present in a *Nannochloropsis gaditana* supernatant was fully retained and associated the result with the fact that chlorophyll was present in lipid droplets that were larger than the membrane cut-off. The results presented in Table 5 also show that the lipids present in the supernatant were highly retained, which might have contributed to the retention of chlorophyll-a.

This is an interesting result since the green color is a major drawback for certain food applications. These results are important not only for the soluble protein fraction enrichment but also for the overall performance of a *Nannochloropsis* biorefinery. The retention of more than 87% of the lipids present in the supernatant, in all the assays, is extremely significant for increasing the overall biorefinery efficiency as the membrane retentate can incorporate the feed stream for a lipid extraction process. The lipids’ retention can be related to the retention of insoluble compounds, such as insoluble proteins, but also to the formation of aggregates between lipids and structural proteins with a mass higher than the membrane molecular weight cut-off [21,42]. 

#### 3.2.3. Fouling and Membrane Cleaning

Membrane fouling during membrane operation is inevitable, and therefore, the easiness and readiness of membrane cleaning is an important feature that needs to be taken into account for industrial membrane operation. The soluble protein mass balance (presented in Section 3.2.2), hydraulic permeance decrease, and resistance (reversible and irreversible) for the different experiments were analyzed to evaluate the possible relationships between the different operating conditions and the easiness/readiness of membrane cleaning. The initial hydraulic permeance Lp_w,i_ (L/(h·m^2^·bar)), experimentally determined before each of the ultrafiltration experiments, and the final hydraulic permeance Lp_w,f_ (L/(h·m^2^·bar)), determined at the end of each experiment, after a hydraulic flush to remove the layer deposited on the membrane surface (reversible fouling), were evaluated and the results are shown in Table 6. The experiments under controlled permeate flux exhibit slightly lower values of hydraulic permeance loss, % Lp_w_ Loss (1 − Lp_w,f_/Lp_w,i_), than the experiments under controlled TMP. The last permeance values of each experiment with *Nannochloropsis* sp. supernatant Lp_f_ (L/(h·m^2^·bar)) are shown in Table 6, confirming that experiments under controlled permeate flux can maintain a higher permeance value until the end of the experiment. On the other hand, comparing the diafiltration operation mode with the concentration operation mode (Table 6), it is notable that after a diafiltration step, the permeance value remains higher than after a concentration step. The difference between the two operation modes becomes even clearer when the operations are performed under controlled permeate flux.

These data indicate that fouling phenomena are more severe in ultrafiltration operated at a concentration mode under controlled TMP, and as a consequence, membrane cleaning might be harder under these conditions. Likewise, the membrane lifetime may be reduced when the ultrafiltration experiments are performed under controlled TMP. Severe fouling during ultrafiltration would occur if the operating conditions selected favored a strong convective transport of foulant toward the membrane surface and the build-up of high local concentrations of these foulants near the membrane surface. In the particular case studied in this work, the presence of biological material, namely proteins and other biological compounds, represents a severe risk of fouling unless adequate operating strategies are adopted (operation under controlled permeate flux). Therefore, diafiltration followed by a concentration step under controlled permeate flux seems to be the best combination of operation conditions, allowing for lower membrane fouling and easier membrane cleaning.

Furthermore, the results obtained for the calculated resistance due to reversible R_rev_ (m^−1^) and irreversible fouling R_irrev_ (m^−1^) for each experiment are shown in Figure 5. Reversible resistance is slightly higher in the experiments under controlled TMP than in the experiments under controlled permeate flux, which is in accordance with other fouling studies [27,43]. This result suggests that it might be easier to clean a membrane, by hydraulic flushing, after ultrafiltration under controlled permeate flux than under controlled TMP. Such a result can be explained by the differences in the deposited layer’s thickness. Vyas et al. [43] reported that the deposited layer formed on the membrane surface during constant permeate flux experiments was very thin. The experiments under controlled TMP exhibit higher values for the irreversible resistance, which might implicate the increase in frequency (and/or intensity) of chemical cleaning cycles, and consequently, the time devoted to cleaning protocols. The membrane lifetime may also be affected by the higher irreversible fouling experienced in the ultrafiltration process under controlled TMP.

## 4. Conclusions

The ultrafiltration process studied, using a 100 kDa MWCO membrane, resulted in obtaining an enriched protein fraction, free from potentially undesirable contaminants. The results show that the ultrafiltration process operated under controlled permeate flux presents significant advantages over operation under controlled transmembrane pressure: higher permeate flux (12 L·m^−2^·h^−1^), higher soluble protein recovery (78% in a diafiltration mode), and lower membrane fouling. This work shows that when working under controlled permeate flux, the operator determines the convective transport conditions of the feed stream toward the surface of the membrane. This means that the operator allows a defined mass flux of feed across the membrane, avoiding the situation that occurs during controlled transmembrane pressure operation where a very high-mass flux of feed might occur during the initial period of operation (when the membrane is still clean and not fouled). As a consequence, a wise choice of permeate flux, imposed and controlled by the operator, allows for working under subcritical (or sustainable) permeate flux conditions, minimizing fouling and thus extending the operation under manageable permeate flux values. The combination of ultrafiltration in a diafiltration mode followed by concentration under controlled permeate flux conditions enhances the soluble protein recovery in the permeate and allows a higher value of volumetric flux to be maintained for the entire diafiltration process. This strategy led to an increase in the average flux (3×) and soluble protein recovery (+34%), along with a decrease in soluble protein rejection (to half of the initial value) and in soluble protein adsorbed to the membrane (to 17% of the initial value). This strategy may be applied to other microalgae in a very straightforward way. It may also be applicable to membrane processes that suffer from significant membrane fouling, e.g., when the feed solution has lipids and other potential foulants.

## Figures and Tables

**Figure 1 membranes-12-00401-f001:**
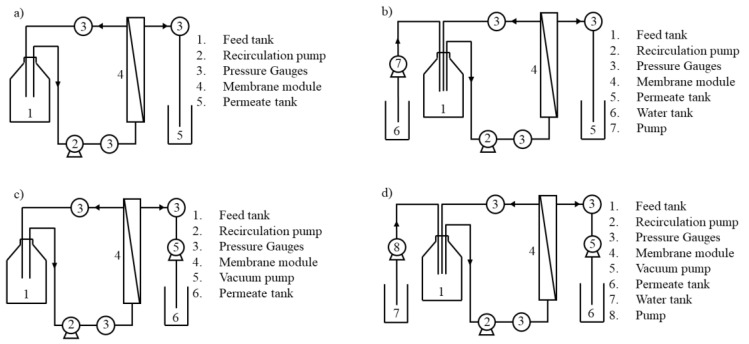
Representation of the ultrafiltration membrane unit during the experiments performed under different operating modes and conditions: (**a**) concentration operation mode, under controlled transmembrane pressure; (**b**) diafiltration operation mode followed by concentration, under controlled transmembrane pressure; (**c**) concentration operation mode, under controlled permeate flux; (**d**) diafiltration operation mode followed by concentration, under controlled permeate flux.

**Figure 2 membranes-12-00401-f002:**
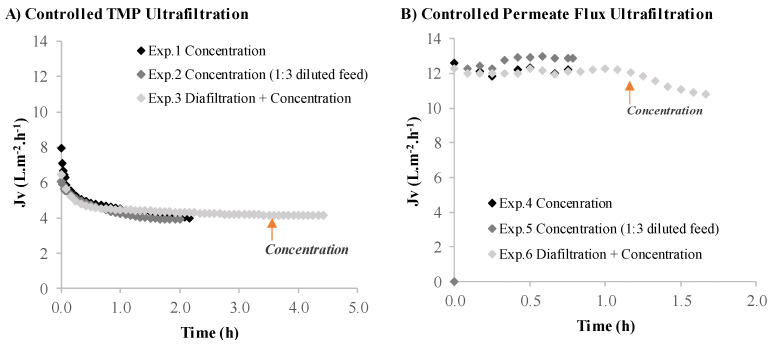
Volumetric flux plotted against operating time during *Nannochloropsis* sp. supernatant ultrafiltration. (**A**) Controlled TMP experiments: Exp.1—ultrafiltration in a concentration operation mode; Exp.2—ultrafiltration in a concentration operation mode using a diluted feed; Exp.3—ultrafiltration in a diafiltration operation mode. (**B**) Controlled permeate flux experiments: Exp.4—ultrafiltration in a concentration operation mode; Exp.5—ultrafiltration in a concentration operation mode using a diluted feed; Exp.6—Ultrafiltration in a diafiltration operation mode.

**Figure 3 membranes-12-00401-f003:**
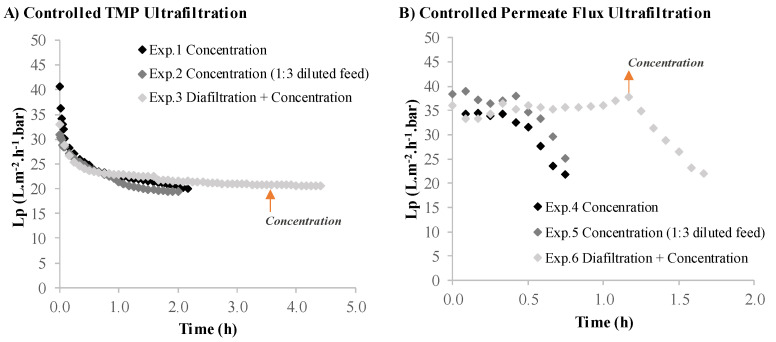
Membrane permeance plotted against operating time during *Nannochloropsis* sp. supernatant ultrafiltration. (**A**) Controlled TMP experiments: Exp.1—ultrafiltration in a concentration operation mode; Exp.2—ultrafiltration in a concentration operation mode using a diluted feed; Exp.3—ultrafiltration in a diafiltration operation mode. (**B**) Controlled permeate flux experiments: Exp.4—ultrafiltration in a concentration operation mode; Exp.5—ultrafiltration in a concentration operation mode using a diluted feed; Exp.6—ultrafiltration in a diafiltration operation mode.

**Figure 4 membranes-12-00401-f004:**
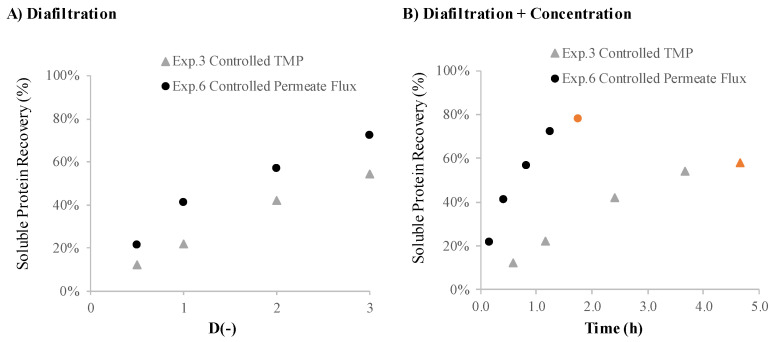
Recovery of *Nannochloropsis* sp. proteins in the permeate, expressed as % (*w*/*w* soluble proteins in the permeate). (**A**) Ultrafiltration in diafiltration mode: protein recovery (%) against the D (-) diafiltration volume; Exp.3—diafiltration under controlled TMP; Exp.6—diafiltration under controlled permeate flux. (**B**) Ultrafiltration in diafiltration mode followed by a concentration step (Exp.3 concentration step (
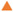
) and Exp.6 concentration step (
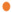
)) with protein recovery (%) against process time (h); Exp.3—diafiltration under controlled TMP; Exp.6—diafiltration under controlled permeate flux.

**Figure 5 membranes-12-00401-f005:**
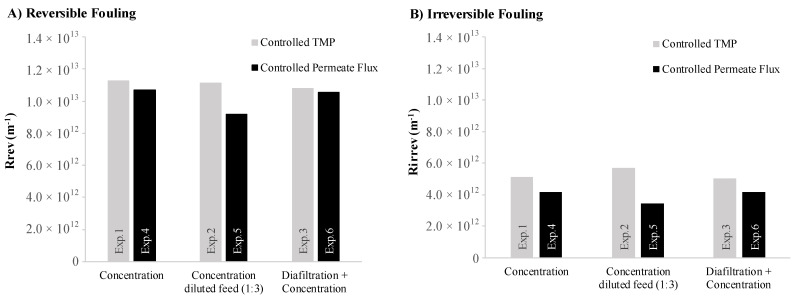
Calculated (**A**) reversible resistance (Rrev) and (**B**) irreversible resistance (Rirrev): Controlled TMP experiments compared with controlled permeate flux experiments. Exp.1/Exp.4—ultrafiltration in a concentration operation mode; Exp.2/Exp.5—ultrafiltration in a concentration operation mode of a diluted feed; Exp.3/Exp.6—ultrafiltration in a diafiltration operation mode.

**Table 1 membranes-12-00401-t001:** Scheme of the ultrafiltration experimental design.

Experiment	Total Feed Concentration (g_SFDW_·L^−1^)	Operation Mode	Operating Conditions	TMP (bar)	*J_v_* (L·m^−2^·h^−1^)
1	30	Concentration	Controlled transmembrane pressure	0.2	
2	10	Concentration		0.2	
3	30	Diafiltration + Concentration		0.2	
4	30	Concentration	Controlled permeate flux		12
5	10	Concentration			12
6	30	Diafiltration + Concentration			12

**Table 2 membranes-12-00401-t002:** Supernatant composition in terms of proteins, lipids, and chlorophyll-a. Results are based on the average of six supernatant samples, used as feed solutions for each membrane experiment.

Total protein (g·L^−1^)	8.77 ± 0.37
Soluble protein (g·L^−1^)	7.28 ± 0.58
Insoluble protein (g·L^−1^)	1.48 ± 0.27
Total lipids (g·L^−1^)	4.37 ± 0.60
Chlorophyll-a (g·L^−1^)	0.08 ± 0.02

**Table 3 membranes-12-00401-t003:** Recovery of *Nannochloropsis* sp. soluble proteins in the permeate and soluble proteins rejection. (**A**) Controlled TMP experiments: Exp.1—ultrafiltration in a concentration operation mode; Exp.2—ultrafiltration in a concentration operation mode of a diluted feed; Exp.3—ultrafiltration in a diafiltration operation mode. (**B**) Controlled permeate flux experiments: Exp.4—ultrafiltration in a concentration operation mode; Exp.5—ultrafiltration in a concentration operation mode of a diluted feed; Exp.6—ultrafiltration in a diafiltration operation mode. Results are based on two replicates for each sample protein analysis. Note: the percentage of soluble protein recovered in the permeate plus the soluble protein rejected (in the retentate) does not reach 100%; this difference is attributed to protein adsorbed by the membrane.

**(A) Controlled TMP Experiments**			
	**Exp.1 Concentration**	**Exp.2 Concentration (1:3 Diluted Feed)**	**Exp.3 Diafiltration + Concentration**
Soluble Protein Recovery (%)	35.0 ± 0.4	40.0 ± 1.5	58.0 ± 2.0
Soluble Protein Rejection (%)	57.0 ± 0.5	50.0 ± 1.8	37.0 ± 2.1
**(B) Controlled Permeate Flux Experiments**			
	**Exp.4 Concentration**	**Exp.5 Concentration (1:3 Diluted Feed)**	**Exp.6 Diafiltration + Concentration**
Soluble Protein Recovery (%)	44.0 ± 4.1	66.0 ± 1.9	78.0 ± 0.4
Soluble Protein Rejection (%)	49.0 ± 4.2	26.0 ± 2.3	19.0 ± 0.4

**Table 4 membranes-12-00401-t004:** Mass balance of soluble proteins. (**A**) Controlled TMP experiments: Exp.1—ultrafiltration in concentration mode; Exp.2—ultrafiltration concentration mode of a diluted feed; Exp.3—ultrafiltration in diafiltration + concentration mode. (**B**) Controlled permeate flux experiments: Exp.4—ultrafiltration in concentration mode; Exp.5—ultrafiltration in concentration mode of a diluted feed; Exp.6—ultrafiltration in diafiltration + concentration mode. Results are based on 2 replicates for each sample protein analysis. ^(^*^)^ calculated through the subtraction of ƩOUT_Permeate+Retentate_ (g) from IN_Feed_ (g).

**(A) Controlled TMP Experiments**			
	**Exp.1 Concentration**	**Exp.2 Concentration (1:3 Diluted Feed)**	**Exp.3 Diafiltration + Concentration**
IN_Feed_ (g)	15.62 ± 0.20	5.40 ± 0.10	7.30 ± 0.06
OUT_Permeate_ (g)	5.49 ± 0.13	2.16 ± 0.12	4.26 ± 0.18
OUT_Retentate_ (g)	3.53 ± 0.05	1.33 ± 0.02	0.34 ± 0.02
ƩOUT_Permeate+Retentate_ (g)	9.02 ± 0.18	3.49 ± 0.14	4.60 ± 0.20
OUT_Adsorbed_ (g) ^(^*^)^	6.60 ± 0.02	1.91 ± 0.04	2.71 ± 0.14
Protein Loss (%)	42.27 ± 0.44	35.37 ± 1.47	37.05 ± 2.01
Proteins Adsorbed (%)	42.27 ± 0.44	35.37 ± 1.47	37.05 ± 2.01
**(B) Controlled Permeate Flux Experiments**			
	**Exp.4 Concentration**	**Exp.5 Concentration (1:3 Diluted Feed)**	**Exp.6 Diafiltration + Concentration**
IN_Feed_ (g)	12.80 ± 0.60	5.52 ± 0.18	10.35 ± 0.03
OUT_Permeate_ (g)	5.62 ± 0.83	3.40 ± 0.01	7.96 ± 0.04
OUT_Retentate_ (g)	2.45 ± 0.03	1.41 ± 0.01	1.40 ± 0.03
ƩOUT_Permeate+Retentate_ (g)	8.07 ± 0.86	4.81 ± 0.01	9.58 ± 0.07
OUT_Adsorbed_ (g) ^(^*^)^	4.73 ± 0.26	0.71 ± 0.17	0.77 ± 0.04
OUT_Adsorbed_ (%)	36.95 ± 2.71	12.93 ± 2.61	7.42± 0.40

**Table 5 membranes-12-00401-t005:** Feed and permeate compositions in terms of insoluble proteins, lipids, and chlorophyll-a. (**A**) Non-diluted feed experiments: under controlled TMP (Exp.1 concentration mode and Exp.3 diafiltration + concentration mode), and under controlled permeate flux (Exp.4 concentration mode and Exp.6 diafiltration + concentration mode). (**B**) Diluted feed experiments: under controlled TMP (Exp.2 concentration mode), and under controlled permeate flux (Exp.5 concentration mode). The results for the permeates from each experiment are based on two replicates for each sample analysis.

**(A) Non-Diluted Feed Experiments**					
		**Permeate**			
		**Controlled TMP**		**Controlled Permeate Flux**	
	**Feed**	**Exp.1 Concentration**	**Exp.3 Diafiltration + Concentration**	**Exp.4 Concentration**	**Exp.6 Diafiltration + Concentration**
Insoluble Proteins (g·L^−1^)	1.48 ± 0.27	ND ^(1)^	ND ^(1)^	ND ^(1)^	ND ^(1)^
Total Lipids (g·L^−1^)	4.37 ± 0.61	0.47 ± 0.05	0.61 ± 0.05	0.47 ± 0.05	0.50 ± 0.05
Chlorophyll-a (g·L^−1^)	0.08 ± 0.02	ND ^(2)^	ND ^(2)^	ND ^(2)^	ND ^(2)^
**(B) Diluted Feed (1:3) Experiments**					
		**Permeate**			
		**Controlled TMP**		**Controlled Permeate Flux**	
	**Feed**	**Exp.2 Concentration**		**Exp.5 Concentration**	
Insoluble Proteins (g·L^−1^)	0.50 ± 0.13	ND ^(1)^		ND ^(1)^	
Total Lipids (g·L^−1^)	1.53 ± 0.13	0.20 ± 0.05		0.18 ± 0.05	
Chlorophyll-a (g·L^−1^)	0.04 ± 0.01	ND ^(2)^		ND ^(2)^	

^(1)^ Detection limit: 5 mg·L^−1^; ^(2)^ Detection limit: 3 µg·L^−1^.

**Table 6 membranes-12-00401-t006:** Initial hydraulic permeance Lp_w,i_, hydraulic permeance after hydraulic flush at the end of each trial Lp_w,f_, hydraulic permeance loss after each experiment Lp_w_ Loss, and the permeance at the end of each *Nannochloropsis* sp. supernatant filtration. (**A**) Controlled TMP experiments and (**B**) Controlled permeate flux experiments. * Permeance value at the end of the diafiltration step and before the concentration step.

**(A) Controlled TMP Experiments**				
	**Exp.1**	**Exp.2**	**Exp.3**	
	**Concentration**	**Concentration (1:3 Diluted Feed)**	**Diafiltration**	**Diafiltration + Concentration**
Lp_w,i_ [L/(h·m^2^·bar)]	219.6	227.3	218.7	218.7
Lp_w,f_ [L/(h·m^2^·bar)]	53.1	49.2	-	53.8
Lp_w_ Loss (%)	76%	78%	-	75%
Lp_f_ [L/(h·m^2^·bar)]	19.9	19.5	21.0 *	20.6
**(B) Controlled Permeate Flux Experiments**				
	**Exp.4**	**Exp.5**	**Exp.6**	
	**Concentration**	**Concentration (1:3 Diluted Feed)**	**Diafiltration**	**Diafiltration + Concentration**
Lp_w,i_ [L/(h·m^2^·bar)]	221.8	219.8	221.7	221.7
Lp_w,f_ [L/(h·m^2^·bar)]	62.1	70.81	-	62.4
Lp_w_ Loss (%)	72%	68%	-	72%
Lp_f_ [L/(h·m^2^·bar)]	21.8	25.2	37.0 *	22.1

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
