# Peer review of "Nannochloropsis sp. Biorefinery: Recovery of Soluble Protein by Membrane Ultrafiltration/Diafiltration"

_membranes, 2022, doi:10.3390/membranes12040401_

Round 1

Reviewer 1 Report

This paper addresses an important problem – define a membrane processing in order to maximize the recovery of the soluble protein fraction from the aqueous supernatant obtained by centrifugation, based on the Nannochloropsis sp. In addition, membrane fouling during operation is studied and operating conditions with minimum fouling are discussed. The structure is concise and clear. Overall, the article is well organized and its presentation is good. However, some issues still need to be improved:

(1) Some paragraphs in the Results and Discussion section mainly lack more in-depth analysis. I recommend author adding several reasoning and comparison through other similar published work. For example, a few paragraphs in Section 3.2.2.

(2) In the conclusions, in addition to summarizing the methods taken and results, please strengthen the explanation of their significance.

(3) I recommend increasing the discussion of the applicability of this study.

Author Response

This paper addresses an important problem – define a membrane processing in order to maximize the recovery of the soluble protein fraction from the aqueous supernatant obtained by centrifugation, based on the Nannochloropsis sp. In addition, membrane fouling during operation is studied and operating conditions with minimum fouling are discussed. The structure is concise and clear. Overall, the article is well organized and its presentation is good. However, some issues still need to be improved:

  • Some paragraphs in the Results and Discussion section mainly lack more in-depth analysis. I recommend author adding several reasoning and comparison through other similar published work. For example, a few paragraphs in Section 3.2.2.

Response 1

The section 3.2.2 was modified, according to the reviewer suggestion. The following text was added:

In Lines 424 to 430: “The results obtained in terms of protein recovery during the diafiltration under controlled TMP are slightly lower than those obtained by Balti et al. [41], during the diafiltration of soluble proteins from Spirulina. The authors reported 64% recovery of soluble proteins in the permeate, using a 150 kDa ceramic membrane. It is also important to note that the author applied two more diavolumes (a total of five) than those applied during the diafiltration described in the present work (a total of three), which might have contributed to a slightly higher protein recovery.”

  1. Balti, R.; Zayoud, N.; Hubert, F.; Beaulieu, L.; Massé, A. Fractionation of Arthrospira platensis (Spirulina) water soluble proteins by membrane diafiltration. Sep. Purif. Technol. 2021, 256, doi:10.1016/j.seppur.2020.117756.

Additionally, in Lines 494 to 500: “The results obtained in terms of chlorophyll-a retention are comparable with those reported by Safi et al. [26], using a 300 kDa PES membrane. The authors reported that the chlorophyll present in a Nannochloropsis gaditana supernatant was fully retained, and associated the result with the fact that chlorophyll was present in lipid droplets that were larger than the membrane cut-off. The results presented in Table 5 also show that the lipids present in the supernatant were highly retained which might have contributed to the retention of chlorophyll-a.”

  1. Safi, C.; Olivieri, G.; Campos, R.P.; Engelen-Smit, N.; Mulder, W.J.; Broek, L.A.M. Van Den; Sijtsma, L. Biorefinery of microalgal soluble proteins by sequential processing and membrane filtration. Bioresour. Technol. 2016, 225, 151–158, doi:10.1016/j.biortech.2016.11.068.
  • In the conclusions, in addition to summarizing the methods taken and results, please strengthen the explanation of their significance.

Response 2

The conclusion section was modified, according to the reviewer suggestion. The following text was added:

In Lines 595 to 604: “This work shows that when working under controlled permeate flux the operator determines the convective transport conditions of the feed stream towards the surface of the membrane. This means that the operator allows a defined mass flux of feed across the membrane, avoiding the situation that occurs during controlled transmembrane pressure operation where a very high mass flux of feed might occur during the initial period of operation (when the membrane is still clean and not fouled). As a consequence, a wise choice of permeate flux, imposed and controlled by the operator, allows for working under sub-critical (or sustainable) permeate flux conditions, minimizing fouling and thus extending operation under manageable permeate flux values”.

  • I recommend increasing the discussion of the applicability of this study.

Response 3

The conclusion section was modified, according to the reviewer suggestion. The following text was added:

In Lines 607 to 612: “Also, this strategy may be applied to other microalgae in a very straightforward way. It may also be applicable to membrane processes that suffer from significant membrane fouling, e.g., when the feed solution contains lipids and other potential foulants.”

Reviewer 2 Report

Dear Editor

Thank you for the invitation to review the manuscript entitles “Nannochloropsis sp. biorefinery: recovery of soluble protein by membrane ultrafiltration/ diafiltration".  This work proposes a way to maximize the potential of a Nannochloropsis sp. biorefinery process, through membrane technology, producing an extract enriched in soluble proteins, free from the insoluble protein fraction, with a low lipid content and eliminating the colored chlorophyll-a. Ultrafiltration operated in a diafiltration mode, under controlled permeate flux conditions, led to the highest soluble protein recovery with the highest constant permeate flux and low membrane fouling.

The results of the manuscript is interested for a wide number of the readers, therefore, I recommend this paper to publish in membrane after major revision. Please see the comments to the authors in the report below.

Best Regards

Comments to the authors:

  • The introduction seems good and covers all aspects of the research.

  • The authors mentioned in the introduction that microalgae protein is incorporated in food products. I recommend authors to provide specific scientific examples of microalgae protein that have been introduced into food products. Moreover, are there any additional applications for these algae-derived proteins?

  • The authors should add brief information on Nannochloropsis sp.
  • The authors discussed of membrane processes very briefly in the introduction; I recommend further information, including membrane types and the benefits and drawbacks of membranes in general.

  • The authors reported in section 2.2 "the biomass has been diluted", more details should be present!

  • What is the purpose of the last step, which includes utilizing ethanol in the clean of the membrane?. I suggest to explain it!

  • In section 3.2.1, the authors should be present a scientific explanation and their opinion on why the volumetric flux value stays higher in the ultrafiltration process under controlled permeate flux (12 L.m-2.h-1).

  • Section 3.2.3 "These data indicate that fouling phenomena are more severe in ultrafiltration operated in a concentration mode under controlled TMP and, as a consequence, membrane cleaning might be harder under these conditions. Likewise, the membrane lifetime may be reduced when the ultrafiltration experiments are performed under controlled TMP." The authors did not present the reason behind the severe effect of fouling phenomenon in ultrafiltration.

  • The conclusion is too short and did not include the significant results found by the authors. The hypothesis of the manuscript should also present in the conclusion.

Author Response

Comments to the authors:

The introduction seems good and covers all aspects of the research. 

  • The authors mentioned in the introduction that microalgae protein is incorporated in food products. I recommend authors to provide specific scientific examples of microalgae protein that have been introduced into food products. Moreover, are there any additional applications for these algae-derived proteins?

Response 1

The introduction section was modified, according to the reviewer suggestion. The following text was added:

In Lines 34 to 37: “Microalgae proteins are a potential suitable source of high-quality protein not only for human food [1], but also for animal feed [2], and in specific cases, such as the case of phycobiliproteins, for nutraceutical and pharmaceutical applications [3].”

  1. Fradique, Ḿonica; Batista, A.P.; Nunes, M.C.; Gouveia, L.; Bandarra, N.M.; Raymundo, A. Incorporation of Chlorella vulgaris and Spirulina maxima biomass in pasta products. Part 1: Preparation and evaluation. J. Sci. Food Agric. 2010, 90, 1656–1664, doi:10.1002/jsfa.3999.
  2. Ansari, F.A.; Guldhe, A.; Gupta, S.K.; Rawat, I.; Bux, F. Improving the feasibility of aquaculture feed by using microalgae. Environ. Sci. Pollut. Res. 2021, 28, 43234–43257, doi:10.1007/s11356-021-14989-x.
  3. Ashaolu, T.J.; Samborska, K.; Lee, C.C.; Tomas, M.; Capanoglu, E.; Tarhan, Ö.; Taze, B.; Jafari, S.M. Phycocyanin, a super functional ingredient from algae; properties, purification characterization, and applications. Int. J. Biol. Macromol. 2021, 193, 2320–2331, doi:10.1016/J.IJBIOMAC.2021.11.064.

In Lines 40 to 42: “Fradique et al. [1], reported to have successfully produced a high protein pasta product through the incorporation of Chlorella vulgaris and Spirulina maxima to semolina flour. The authors also reported that the cooking quality of pastas was not compromised.”

  1. Fradique, Ḿonica; Batista, A.P.; Nunes, M.C.; Gouveia, L.; Bandarra, N.M.; Raymundo, A. Incorporation of Chlorella vulgaris and Spirulina maxima biomass in pasta products. Part 1: Preparation and evaluation. J. Sci. Food Agric. 2010, 90, 1656–1664, doi:10.1002/jsfa.3999.

In Lines 47 to 51: “Qazi et al. [8], studied the impact on dough rheology and bread quality after the enrichment with an ethanol pre-treated Tetraselmis chuii protein fraction, in order to remove the green color and improve the sensory properties. However, besides the clear improvement in dough rheology and bread quality, the authors reported that the green pigmentation was not fully eliminated.”

  1. Qazi, W.M.; Ballance, S.; Uhlen, A.K.; Kousoulaki, K.; Haugen, J.E.; Rieder, A. Protein enrichment of wheat bread with the marine green microalgae Tetraselmis chuii – Impact on dough rheology and bread quality. Lwt 2021, 143, 111115, doi:10.1016/j.lwt.2021.111115.

  • The authors should add brief information on Nannochloropsis sp.

Response 2

The introduction section was modified, according to the reviewer suggestion. The following text was added:

In Lines 61 to 70: “Nannochloropsis sp. is a small marine microalga (Eustigmatophyceae). The cells of Nannochloropsis sp. are small (3-5 µm) spherical or sightly ovoid [14], with a single chloroplast occupying a large volume of the cell [15], and with a strong cell wall composed of a bilayer structure consisting of a cellulosic inner wall and a hydrophobic algaenan outer layer [16,17]. Algaenan is a resistant biopolymer, and the algaenan layer has been described to be composed of long, straight-chain, saturated aliphatics with ether cross-links [16,17]. In numerous cases, algaenan is part of a layer called trilaminar layer structure (TLS), which exhibits two high electron density outside layers sandwiching one layer with low electron density [18,19]. The resistance of the algaenan cell wall and the small cell diameter confers high mechanical robustness to Nannochloropsis sp. cells.”

  1. HIBBERD, D.J. Notes on the taxonomy and nomenclature of the algal classes Eustigmatophyceae and Tribophyceae (synonym Xanthophyceae). Bot. J. Linn. Soc. 1981, 82, 93–119, doi:10.1111/j.1095-8339.1981.tb00954.x.
  2. Cecchin, M.; Berteotti, S.; Paltrinieri, S.; Vigliante, I.; Iadarola, B.; Giovannone, B.; Maffei, M.E.; Delledonne, M.; Ballottari, M. Improved lipid productivity in Nannochloropsis gaditana in nitrogen-replete conditions by selection of pale green mutants. Biotechnol. Biofuels 2020, 13, 1–14, doi:10.1186/s13068-020-01718-8.
  3. Chua, E.T.; Schenk, P.M. A biorefinery for Nannochloropsis: Induction, harvesting, and extraction of EPA-rich oil and high-value protein. Bioresour. Technol. 2017, 244, doi:10.1016/j.biortech.2017.05.124.
  4. Scholz, M.J.; Weiss, T.L.; Jinkerson, R.E.; Jing, J.; Roth, R.; Goodenough, U.; Posewitz, M.C.; Gerken, H.G. Ultrastructure and Composition of the Nannochloropsis gaditana Cell Wall. Eukaryot. Cell 2014, 13, 1450–1464, doi:10.1128/EC.00183-14.
  5. Dunker, S.; Wilhelm, C. Cell wall structure of coccoid green algae as an important trade-offbetween biotic interference mechanisms and multidimensional cell growth. Front. Microbiol. 2018, 9, doi:10.3389/fmicb.2018.00719.
  6. Allard, B.; Templier, J. Comparison of neutral lipid profile of various trilaminar outer cell wall (TLS)-containing microalgae with emphasis on algaenan occurrence. Phytochemistry 2000, 54, 369–380, doi:10.1016/S0031-9422(00)00135-7.

  • The authors discussed of membrane processes very briefly in the introduction; I recommend further information, including membrane types and the benefits and drawbacks of membranes in general.

Response 3

The introduction section was modified, according to the reviewer suggestion. The following text was added:

In Lines 88 to 94: "Membrane processes, namely the commonly used ultrafiltration or nanofiltration are pressure driven processes. The selective transport of the different compounds of a solution is mainly regulated by molecular size exclusion mechanisms, but also by the Coulombic and hydrophobic interactions between the compounds present and the membrane selective top layer [24]. Generally, membrane technologies are economic, easy to operate and to scale-up, although not providing high product selectivity when compared e.g., with chromatography processes.”

  1. Crespo, J.G.; Brazinha, C. Membrane processing: Natural antioxidants from winemaking by-products. Filtr. Sep. 2010, 47, 32–35, doi:10.1016/S0015-1882(10)70079-3.

  • The authors reported in section 2.2 "the biomass has been diluted", more details should be present!

Response 4

The section 2.2 was modified, according to the reviewer suggestion. The following text was added:

In Line 162 to 166: “The supernatant (30 gSFDW.L-1 and 60 gDW.L-1) from the centrifugation of Nannochloropsis sp. disrupted biomass was diluted with Nannochloropsis sp. culture medium, which has a salt concentration of 30 gDW.L-1, to a dilution ratio of 1:3. After the dilution, the concentration of the diluted supernatant was 10 gSFDW.L-1 and 40 gDW.L-1.”

  • What is the purpose of the last step, which includes utilizing ethanol in the clean of the membrane? I suggest to explain it!

Response 5

The section 2.3 was modified. The following explanation was added:

In Line 291 to 292: “the last step consisted of circulating ethanol (at 45 ± 2 °C) for 20 min, with a concentration of 70% (v/v), to recover the hydrophilicity of the membrane surface and restore the water permeability [33].”

  1. Tian, J. yu; Chen, Z. lin; Yang, Y. ling; Liang, H.; Nan, J.; Li, G. bai Consecutive chemical cleaning of fouled PVC membrane using NaOH and ethanol during ultrafiltration of river water. Water Res. 2010, 44, 59–68, doi:10.1016/J.WATRES.2009.08.053.

  • In section 3.2.1, the authors should be present a scientific explanation and their opinion on why the volumetric flux value stays higher in the ultrafiltration process under controlled permeate flux (12 L.m-2.h-1).

Response 6

We decided to add this paragraph to the conclusions section, lines 595 to 604, in order to emphasize the relevance of operation under controlled permeate flux.

“When working under controlled permeate flux the operator determines the convective transport conditions of the feed stream towards the surface of the membrane. This means that the operator allows a defined mass flux of feed across the membrane, avoiding the situation that occurs during controlled transmembrane pressure operation where a very high mass flux of feed might occur during the initial period of operation (when the membrane is still clean and not fouled). As a consequence, a wise choice of permeate flux, imposed and controlled by the operator, allows for working under sub-critical (or sustainable) permeate flux conditions, minimizing fouling and thus extending operation under manageable permeate flux values.”

  • Section 3.2.3 "These data indicate that fouling phenomena are more severe in ultrafiltration operated in a concentration mode under controlled TMP and, as a consequence, membrane cleaning might be harder under these conditions. Likewise, the membrane lifetime may be reduced when the ultrafiltration experiments are performed under controlled TMP." The authors did not present the reason behind the severe effect of fouling phenomenon in ultrafiltration.

Response 7

The section 3.2.3. was modified, according to the reviewer suggestion. The following text was added:

In Lines 552 to 557: “Severe fouling during ultrafiltration will occur if the operating conditions selected favors a strong convective transport of foulants towards the membrane surface and the build-up of high local concentrations of these foulants near the membrane surface. In the particular case studied in this work, the presence of biological material, namely proteins and other biological compounds, represents a severe risk of fouling unless adequate operating strategies are adopted (operation under controlled permeate flux).”

  • The conclusion is too short and did not include the significant results found by the authors. The hypothesis of the manuscript should also present in the conclusion.

Response 8

Taking into account the Reviewer’s comment, the following sentences were added to the conclusions section:

In Lines 595 to 604: “This work shows that when working under controlled permeate flux the operator determines the convective transport conditions of the feed stream towards the surface of the membrane. This means that the operator allows a defined mass flux of feed across the membrane, avoiding the situation that occurs during controlled transmembrane pressure operation where a very high mass flux of feed might occur during the initial period of operation (when the membrane is still clean and not fouled). As a consequence, a wise choice of permeate flux, imposed and controlled by the operator, allows for working under sub-critical (or sustainable) permeate flux conditions, minimizing fouling and thus extending operation under manageable permeate flux values.”

In Lines 607 to 609: This strategy led to an increase of the average flux (3x) and soluble protein recovery (+34%), a decrease of soluble protein rejection (to half of the initial value) and of soluble protein adsorbed to the membrane (to 17% of the initial value).

In Lines 609 to 612: Also, this strategy may be applied to other microalgae in a very straightforward way. It may also be applicable to any membrane processing with significant membrane fouling, e.g., when the feed solution has lipids and other potential foulants.”

Round 2

Reviewer 2 Report

I would like to inform you that the authors answer all our comments, therefore I accept the manuscript to publish in Membranes